# Effects of different parameters of Tai Chi on the intervention of chronic low back pain: A meta-analysis

Hailun Kang[1], Min Yang[1]*, Mengke Li[1], Rui Xi[2], Qin Sun[3], Qinqin Lin[1]

1 School of Physical Education, Yanshan University, Qinhuangdao, China, 2 Sports Rehabilitation Center, China Institute of Sport Science, Beijing, China, 3 Department of PE, Beijing Wuzi University, Beijing, China

* yangmin@ysu.edu.cn

## Abstract

### Objective

To evaluate the effects of Tai Chi in the treatment of patients with chronic low back pain by Meta-analysis and to investigate its influencing factors.

### Methods

The study searched eight databases (PubMed, Embase, The Cochrane Library, Web of Science, China Knowledge Network, Wanfang, VIP, and CBM) from inception to October 2023. Two investigators independently selected 10 eligible randomized controlled trials (RCT) against inclusion and exclusion criteria, followed by data extraction and study quality assessment by ROB 2. The outcomes of interest were pain intensity and disability. The studies were combined using meta-analysis when statistical pooling of data was possible. The quality of the evidence was assessed using the GRADE approach.

### Results

10 randomized controlled studies with a total sample of 886 cases were included, of which 4 (40%) were assessed as low risk of bias. The effect size of Tai Chi for chronic low back pain was [Weighted Mean Difference (*WMD*) with 95% Confidence Interval (CI) = -1.09 (-1.26, -0.92), $p < 0.01$], all achieving large effect sizes and statistically significant; the effect size for disability was [Standard Mean Difference (*SMD*) with 95% CI = -1.75 (-2.02, -1.48), $p < 0.01$], and the combined effect sizes of physical health and mental health for quality of life were [*WMD* (95% CI) = 4.18 (3.41, 4.95), $p < 0.01$; *WMD* (95% CI) = 3.23 (2.42, 4.04), $p < 0.01$] respectively. The incidence of adverse reactions was low. Meta regression and subgroup analysis showed that there was no significant effect on intervention measures (Tai Chi alone, Tai Chi as additional therapy, water Tai Chi), Tai Chi school (Chen and Yang) and the number of total intervention sessions (> 30 and ≤ 30). The evidence quality evaluation showed that the evidence of pain, physical health of quality of life and mental health score was medium quality, while the evidence of disability and adverse reactions was low quality.

**Data Availability Statement:** All relevant data are within the manuscript and its Supporting Information files.

**Funding:** The funder L. QQ. played a role in the preparation of the manuscript.

**Competing interests:** The authors have declared that no competing interests exist.

## Conclusions

Tai Chi has an obvious effect of in relieving chronic low back pain. Tai Chi alone and Tai Chi as supplementary therapy have good effects. Tai Chi in water have not been verified. Chen style Tai Chi and Yang's Tai Chi, intervention more than 30 times or less than 30 times had no significant difference in the effect of intervention on CLBP.

## 1 Introduction

Chronic Low Back Pain (CLBP) refers to skeletal muscle pain that extends from the 12th rib to the hip fold for more than 3 months, with or without leg pain (sciatica) [1, 2]. CLBP is a globally prevalent public health problem with some risk of disability [3]. Pain is the main symptom of low back pain, and severe low back pain can lead to a significant reduction in quality of life and employment rates. Compared with conventional bed rest, therapeutic rehabilitation based on active training has a significant improvement in the subjective perception of CLBP [4]. The exercise therapies which are commonly used and widely recognized include McKenzie therapy, core stability training, etc., which improve the functional limitation of CLBP patients and relieve low back pain by enhancing lumbar muscle stability and endurance, and improving lumbar spine flexibility [5].

Traditional Chinese aerobic fitness exercise Tai Chi is a form of exercise of low and moderate intensity. It has gradually been popularized due to the advantages of the combination with mindfulness meditation, exercising through slow overall movement to improve musculoskeletal strength and joint stability, and it is mostly used for disease prevention and management. Tai Chi has the advantage of practicing physical and mental exercise, and the evidence of medium and high quality supports that Tai Chi exercise is more effective than passive control in 14 chronic diseases [6]. In China, Tai Chi has produced many schools on the basis of the same boxing theory, and there are currently six major schools with Chen-style, Yang-style, Sun-style, Hao-style, Wu-style, and He-style as the mainstream. These schools have their own characteristics, such as the ups and downs of Chen-style, the neat stretching of Yang-style, the smooth movements of Wu-style, the rigorous elegance of Hao-style, the opening and closing of Sun-style, and the step-and-loop circle of He-style [7], but it is unknown whether there are differences in the application of different styles of Tai Chi. Ai Chi is a new "physical-mental" treatment created by Japanese medical expert Jun Konno based on Tai Chi, and Qigong. With the rise of water sports, based on oriental traditional medicine theory, the rehabilitation method represented by Ai Chi has become one of the mainstream water rehabilitation technology in the world [8, 9]. The 4-week Ai Chi treatment program significantly improved dysfunction and overall core muscle endurance in patients with CLBP, and it seems to have the added benefit of improving one-legged standing balance [10].

In recent years, a review of Tai Chi intervention CLBP published found that previous studies were limited in the inclusion of literature, possibly because the number of original studies on Tai Chi intervention was relatively small, and the studies mostly included Qigong or yoga for common comparison [11, 12]. And there is no comprehensive research on Ai Chi at present. Based on the above research, this study included three kinds of Tai Chi interventions (Tai Chi alone, Tai Chi as an add-on therapy and Ai Chi) for quantitative analysis, and explored some influencing factors of the effectiveness of Tai Chi intervention in CLBP.

## 2 Data source and methods

### 2.1 Literature registration

A meta-analysis of the scientific literature was conducted according to the guidelines for systematic review and meta-analysis (PRISMA). The review is registered in the International Registry of Systematic Reviews (Prospero) under the identification number CRD42023427198.

### 2.2 Literature retrieval strategy

A systematic search of the literature prior to October 2023 was performed using the databases (PubMed, Embase, The Cochrane Library, Web of Science, China Knowledge Network, Wanfang, VIP, and CBM) to collect randomized controlled trials (RCTs) of Tai Chi for the treatment of CLBP, with the search terms such as "Low Back Pain", "Lumbago", "Low Back Ache", "Low Backache", "LBP", "Tai Ji", "Tai Chi", """randomized controlled trial", "RCT", etc. In addition, references from the studies were retrospectively included to supplement access to relevant literature. Take PubMed as an example and see Box 1 for the detailed search strategy.

### Box 1. PubMed search strategy

#1 Low Back Pain "[Mesh]

#2 Back Pain, Low OR Back Pains, Low OR Low Back Pains OR Pain, Low Back OR Pains, Low Back OR Lumbago OR Lower Back Pain OR Back Pain, Lower OR Back Pains, Lower OR Lower Back Pains OR Pain, Lower Back OR Pains, Lower Back OR Low Back Ache OR Ache, Low Back OR Aches, Low Back OR Back Ache, Low OR Back Aches, Low OR Low Back Aches OR Low Backache OR Backache, Low OR Backaches, Low OR Low Backaches OR Low Back Pain, Postural OR Postural Low Back Pain OR Low Back Pain, Posterior Compartment OR Low Back Pain, Recurrent OR Recurrent Low Back Pain OR Low Back Pain, Mechanical OR Mechanical Low Back Pain

#3 #1 OR #2

#4 Tai Ji "[Mesh]

#5 Tai-ji OR Tai Chi OR Chi, Tai OR Tai Ji Quan OR Ji Quan, Tai OR Quan, Tai Ji OR Taiji OR Taijiquan OR T'ai Chi OR Tai Chi Chuan

#6 #4 OR #5

#7 randomized controlled trial[Publication Type] OR randomized OR placebo

#8 #3 AND #6 AND #7

### 2.3 Literature screening

Inclusion criteria: 1. The study subjects should meet the diagnostic criteria for CLBP. Age $\geq$ 18 years old, gender, race, nationality, course of disease are not limited. 2. Intervention measures: (1) the comparison among Tai Chi, Ai Chi and the control group (routine care, blank control, placebo); (2) the comparison between Tai Chi as an add-on therapy combined with the other treatments and other treatments alone (the other treatment is the same). The intervention included different forms of Tai Chi, such as Tai Chi alone, Tai Chi in combination with other usual treatments, or Ai Chi. 3. The primary outcome was the assessment of low

back pain. Visual Analogue Scale (VAS) and Numerical Rating Scale (NRS) are usually used to report the results of low back pain, and the values of the two scales represent similar meanings and are comparable. Secondary outcomes were assessment of the lumbar disability scale, quality of life, and adverse events. 4. The type of study was randomized controlled trials (RCTs). Only Chinese and English language literature were included.

Exclusion criteria: 1. Systematic appraisal or secondary analysis. 2. Inconsistent outcome indicators. 3. Incomplete data. 4. Repetitive literature.

## 2.4 Literature screening and data extraction

Two evaluators independently searched the studies. Firstly, those studies meeting the research purpose were selected by reading the title and abstract. Then, after the full texts were carefully read, the final screening and review were conducted according to the inclusion and exclusion criteria. Meanwhile, we also cross-checked, and extracted the data uniformly, with the disagreements resolved by discussion and group review. Data extraction included authors, publication years, sample size, age, intervention measures, total intervention sessions, outcome indicators and etc.

## 2.5 Statistical methods

Meta-analysis of the data was performed using Review Manager 5.4.1 and Stata 16 software. The outcome indexes of this study were continuous variable indexes; the weighted mean difference (WMD) and the standard mean difference (SMD) were used to represent the combined results with 95% confidence interval (CI), and the combined effect size was statistically significant when Q test p $\leq$ 0.05 [13]; According to the effect value reference standard advocated by Cohen, the absolute value of the effect size was trivial (WMD < 0.2), small (0.2 < WMD < 0.5), moderate (0.5 < WMD < 0.8), and large (WMD > 0.8) [14]; Heterogeneity of included studies was determined by the $I^2$ test, with $I^2$ values of 25%, 50%, and 75% representing low, moderate, and high heterogeneity, respectively. When $I^2 < 50\%$ and P > 0.1, indicating that the heterogeneity among studies was low, and the effect size was combined using the fixed effect model. When $I^2 > 50\%$ and P < 0.1, a random-effects model was used to analyze the source of heterogeneity by subgroup, Meta regression analysis and subgroup analysis. Sensitivity analysis was performed by removing the included studies one by one, and the results were robust if the heterogeneity did not significantly reduce. The analysis of meta regression and subgroup were conducted according to intervention measures (Tai Chi alone, Tai Chi as an add-on therapy and Ai Chi), the styles of Tai Chi (Yang's, Chen's and other styles), and the total intervention sessions ($> 30$ times and $\leq 30$ times). Publication bias was evaluated using a comparative funnel plot and a Egger test.

## 2.6 Literature quality and evidence gradation assessment

The quality assessment of the included studies used the Parallel Design Trial Risk of Bias Assessment Tool (ROB 2) recommended by the Cochrane Handbook, which covered five domains: randomization process, deviation from established interventions, missing outcome data, outcome measurement, and outcome selection reporting, and assessed the overall risk of bias of the outcomes reported by individual studies based on the evaluation of several signal questions within each domain [15]. Grading of Recommendations-assessment, Development and Evaluation (GRADE) is a rating system proposed by the GRADE Working Group. The initial grade of the evidence for RCTs was of high quality, downgraded according to any of the RCTs' risk of bias, inconsistency, indirectness, publication bias, or precision, with the grade of the evidence indicated as high, moderate, low, or very low [16]. Two researchers

independently assessed the grade of evidence, with dissenting opinions discussed by a third one and then jointly decided.

## 3 Results

### 3.1 Literature screening results

According to the formulated search strategy, 272 studies articles were initially generated from the identification stage. A total of 31 studies were deemed suitable for inclusion after screening for duplicates literature and reading the title, abstract, and full text of the papers through End-Note X9 literature management software. Among them, some studies were excluded, including that four study was inconsistent with the intervention, seven studies were inconsistent with the outcome index, two studies were inconsistent with the experimental design, one study conducted the repetitive research object, and 3 research programs and three studies with the repetitive data. The research finally included ten studies to conduct meta-analysis [17–26]. The literature screening process and results are shown in Fig 1.

### 3.2 Basic characteristics of the included literature

Of the 10 included literature, 6 studies were in Chinese and four were in English; One study was conducted in Australia and the remaining nine were all completed in China; A total of 886 patients were involved, with the average age of 29.06 ± 9.53 to 65.10 ± 3.57 years. The 10 randomized controlled trials included 6 studies [17–19, 21, 23, 25] with Tai Chi intervention alone, 3 studies [20, 22, 26] received Tai Chi as an add-on therapy in combination with other treatments (such as massage, acupuncture and other conventional physical therapy), and 1 study [24] with Ai Chi intervention. In the selection of Tai Chi school, 3 studies [18, 19, 23] chose Chen-style Tai Chi, 2 studies [21, 25] chose Yang-style Tai Chi, other studies do not specify. The total number of interventions is calculated by multiplying the intervention duration and the intervention frequency. In terms of total intervention sessions, these studies were divided into > 30 times (5 studies [19, 21–23, 25]) and ≤ 30 times (5 studies [17, 18, 20, 24, 26]). Among them, there was 1 study referring to adverse events and 5 studies with research grants [17, 20, 22, 23, 26]. The basic characteristics of the included studies are presented in Table 1.

### 3.3 Quality evaluation of the included literature

The 10 included RCTs were assessed against the five domains of the risk of bias assessment tool ROB2.0. One RCT [25] was considered to be at high risk and 6 RCTs [17–19, 22, 24, 26] were potentially at risk. Fig 2(A) and 2(B) showed the details of the assessment. In the assessment of risk of bias for the randomized process, 5 RCTs [18, 19, 22, 24, 26] were rated as potentially at risk because they did not describe whether the allocation sequence was hidden. In the assessment of deviation from the intervention, 4 RCTs [20–23] were at low risk of bias and 5 RCTs [17–19, 24, 26] were at possible risk of bias. One study [25] had a more severe intervention deviation and was assessed as high risk of bias. 10 studies [17–26] were not blinded for this type of study could not be blinded to subjects and intervention implementers. No significant risk bias was found in the assessment of missing outcome data, outcome measures and outcome selection reporting.

### 3.4 Meta-analysis results

**3.4.1 Effect of Tai Chi on primary outcome indicators.** As the included studies used different pain assessment scales (10 studies used either a 0 to 10 mm VAS or a 0 to 10 NRS and 2

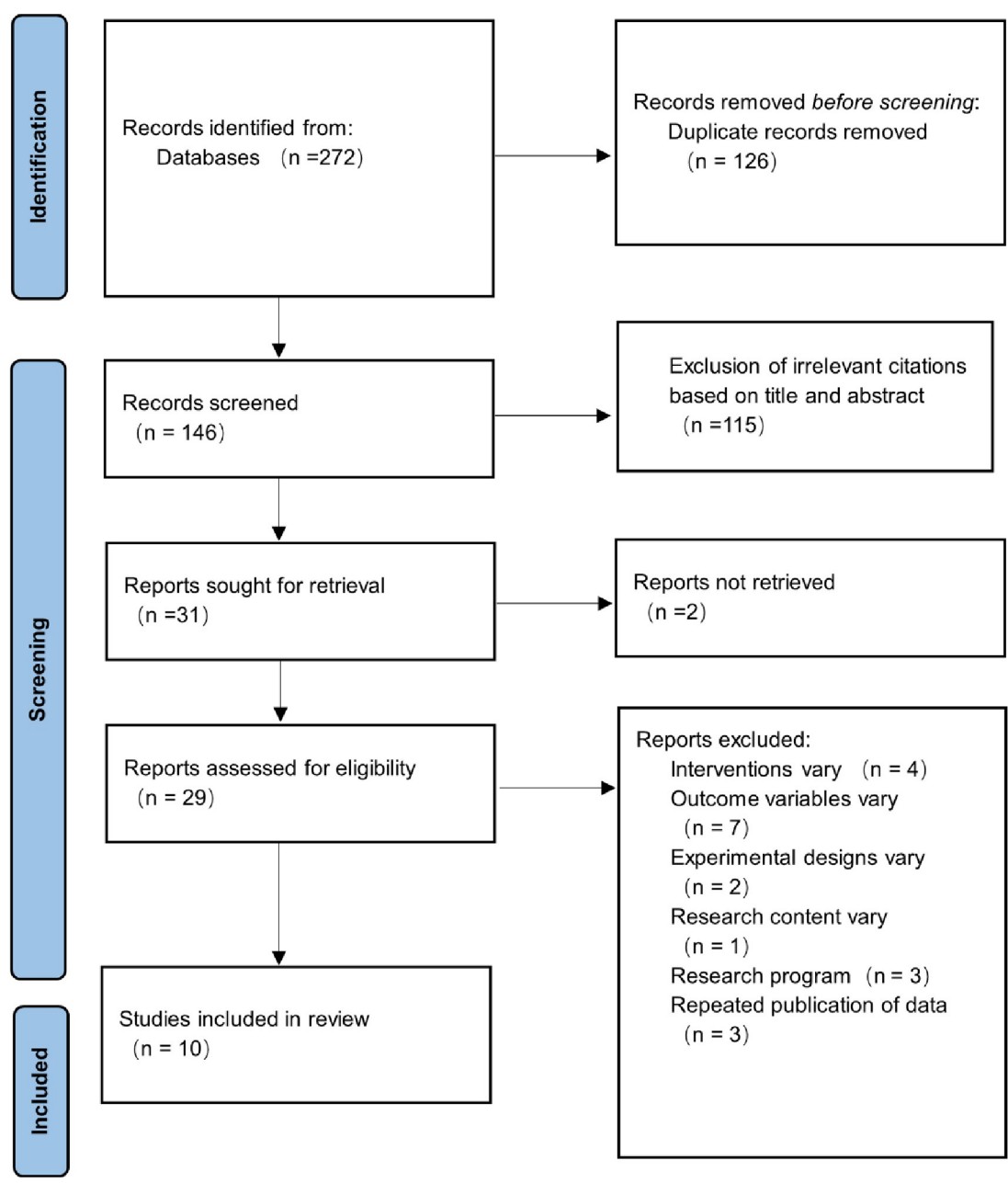

**Fig 1. Literature screening process.**

studies used a 0 to 100 mm VAS), all scales have been uniformly converted to the scale of 0 to 10 grades to facilitate statistical analysis. The 10 studies included were tested for heterogeneity and the results are shown in Fig 3(A). it showed that there was high heterogenicity ($p = 0.000 < 0.1$, $I^2 = 86\%$) among the included studies. Therefore, the random effects model was used to analyze the heterogeneity among the multiple data groups in this meta-analysis, which reflected the possibility of potential regulatory variables. The results of the meta-analysis showed that the combined effect size of Tai Chi on low back pain was [$WMD$ (95% CI) = -1.09 (-1.26, -0.92), $p < 0.01$], reaching the level of large effect. Sensitivity analysis by removing

**Table 1. Basic characteristics of the included studies.**

| Authors (year) | Region (language) | Sample size (T/C) | Average age (T/C years old) | Low back pain type | Interventions | | School | Duration; time; frequency (week; minute; times/ week) | Index | Adverse events | Grant |
|---|---|---|---|---|---|---|---|---|---|---|---|
| | | | | | **T** | **C** | | | | | |
| Hall *et al* [17] 2011 | Australia (English) | 80/80 | 43.4±13.5/ 44.3 ±13.0 | Chronic non-specific low back pain | Tai Chi | Routine care | NR | 10/40/2 | NRS, RMDQ | Yes | Yes |
| Muharram *et al* [18] 2011 | China (English) | 82/82 | 43.6/ 43.3 | Chronic low back pain | Chen-style Tai Chi | Routine care | Chen | 4/60/6 | VAS, SF-36 | NR | NR |
| Wu *et al* [19] 2013 | China (English) | 141/47 | 37.5 ± 5.2/ 38.2 ± 5.8 | Chronic non-specific low back pain | Chen-style Tai Chi | Blank control | Chen | 12/45/5 | VAS | NR | NR |
| Lin *et al* [20] 2015 | China (Chinese) | 30/30 | 41.00±9.22/ 41.00±9.22 | Chronic low back pain | Cloud hands +Massage | Massage | NR | 2/Not calculated/ 6 | VAS | NR | Yes |
| Lu *et al* [21] 2017 | China (Chinese) | 54/54 | 63.37±6.59/ 61.97±6.00 | Chronic non-specific low back pain | Yang-style Tai Chi | Routine care | Yang | 12/40/5 | VAS, SF-36 | NR | NR |
| Fan [22] 2018 | China (Chinese) | 20/20 | 56.4±9.12/ 55.7 ±8.64 | Chronic non-specific low back pain | Tai Chi +Moxibustion | Moxibustion | NR | 12/70/7 | VAS | NR | Yes |
| Liu *et al* [23] 2019 | China (English) | 15/15 | 58.13 ± 5.38/ 58.4 ± 5.08 | Chronic non-specific low back pain | Chen-style Tai Chi | Blank control | Chen | 12/60/3 | VAS | NR | Yes |
| Wang [24] 2020 | China (Chinese) | 18/18 | 31.06±8.76/ 29.06±9.53 | Chronic non-specific low back pain | Ai Chi | physical therapy | NR | 6/45/3 | NRS, RMDQ, SF-36 | NR | NR |
| Wang [25] 2021 | China (Chinese) | 10/10 | 65.10±3.57/ 62.30±3.65 | Chronic low back pain | Yang-style Tai Chi | Routine care | Yang | 12/60/3 | NRS, RMDQ, ODI | NR | NR |
| Shen *et al* [26] 2021 | China (Chinese) | 40/40 | 36.00±11.39/ 35.38±8.76 | Chronic non-specific low back pain | Standing meditation +Massage | Massage | NR | 2/Not calculated/ 7 | VAS, ODI | NR | Yes |

T: test group, C: control group; NRS: numerical rating scale, VAS: visual analogue scale, ODI: The Oswestry Disability Index, RMDQ: Roland-Morris Disability Questionnaire; NR: not reported.

each study individually showed that excluding any one study had no significant effect on the overall effect size, indicating that the meta-analysis was robust, as shown in Fig 3(B).

The results of meta-analysis showed high heterogeneity among individual studies. This study will further explore the influence of the above three variables on the intervention effect through meta regression analysis and subgroup analysis from three aspects of different intervention measures, Tai Chi schools and practice parameters, as shown in Table 2.

Table 3 subgroup analysis results showed that the effect of Tai Chi alone on reducing the pain in CLBP patients was better than that of routine nursing, placebo and blank control group [WMD (95% CI) = -1.23 (-1.46, -0.99), p < 0.01; $I^2$ = 86%]; and the effect of Tai Chi combined with other interventions (acupuncture, massage) on relieving pain in CBLT patients was better than the control group of routine treatments [WMD (95% CI) = -1.02 (-1.29, -0.75),

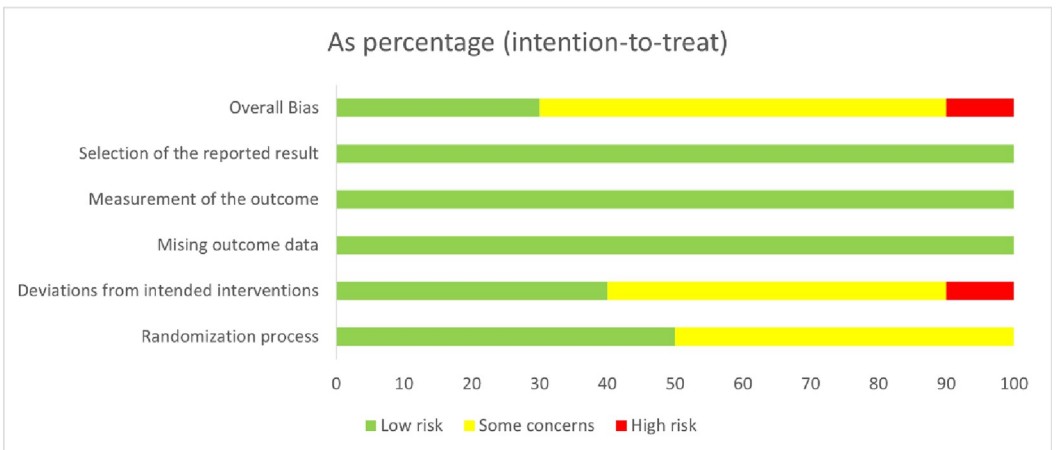

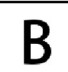

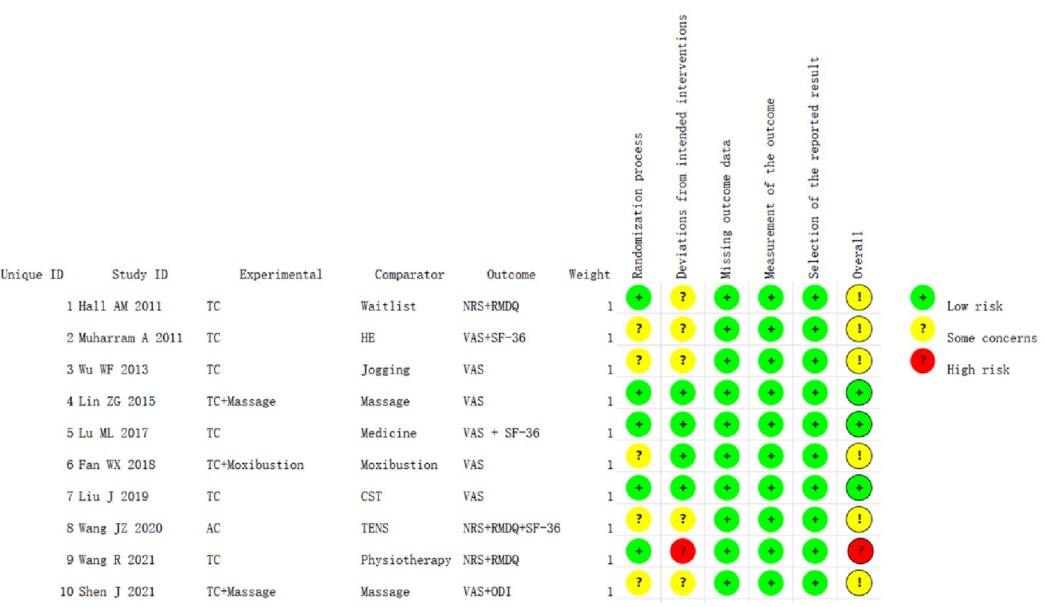

**Fig 2.** (A) Risk bias plot of included studies; (B) Summary of risk bias in the included studies.

p < 0.01; $I^2$ = 83%]; and just one study was included in Ai Chi research [WMD (95% CI) = -0.54 (-0.86, -0.22), p < 0.01]. The statistical differences between subgroups was significant [WMD (95% CI) = -0.54 (-.86, -0.22), p < 0.01].

Among the included studies, 5 of them clearly pointed out the schools of Tai Chi, including 3 of Chen's and 2 of Yang's. There was no statistical difference between the two subgroups (p = 0.42; $I^2$ = 0%): Chen's [WMD (95% CI) = -1.37 (-1.90, -0.84), p<0.01; $I^2$ = 91%], Yang's

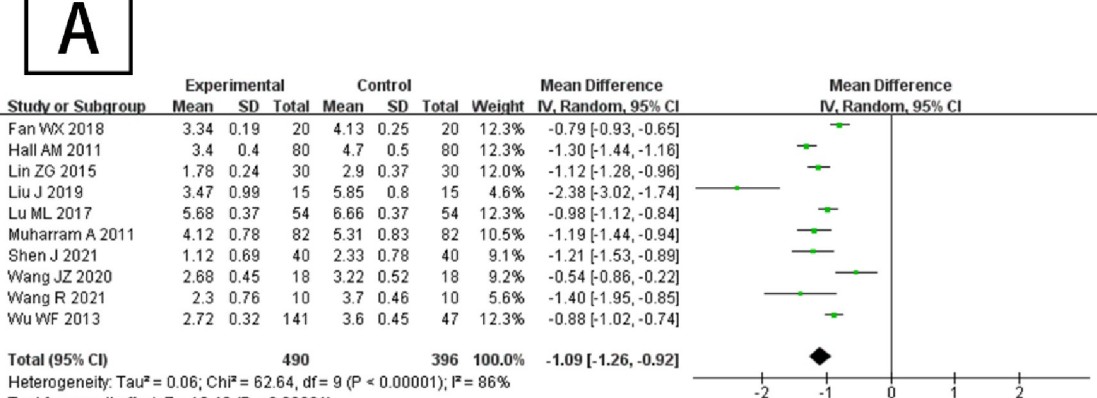

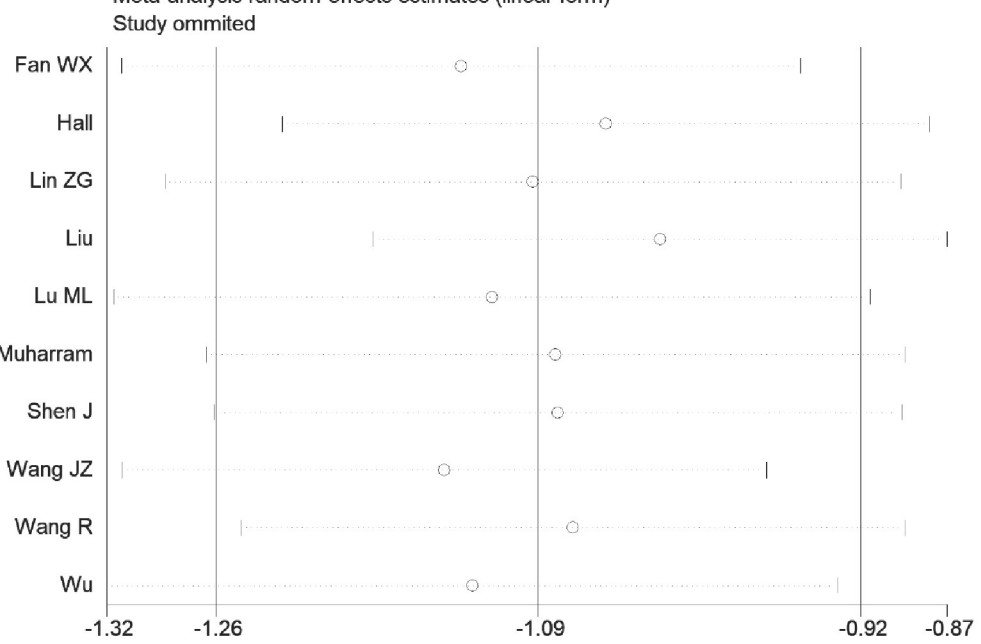

**Fig 3.** (A) Meta-analysis forest plot of the effect of Tai Chi on pain relief in CLBP patients; (B) Sensitivity analysis.

[WMD (95% CI) = -1.10 (-1.48, -0.73), p < 0.01; $I^2$ = 52%], and the others [WMD (95% CI) = -1.00 (-1.26, -0.75), p < 0.01; $I^2$ = 89%]. Subgroups of total number of exercises showed that the difference between groups was not statistically significant (p = 0.96; $I^2$ = 0%). Tai Chi training more than 30 times can improve the pain symptoms of CLBP patients [WMD (95% CI) = -1.09 (-1.33, -0.84), p < 0.01; $I^2$ = 85%]; while training less than 30 times was also able to improve the pain symptoms of CLBP patients [WMD (95% CI) = -1.10(-1.31, -0.88), p < 0.01; $I^2$ = 79%].

**Table 2. Meta-regression analysis and subgroup analysis of tai chi on pain intensity.**

| Variant | | | Correlation coefficient | 95%CI | P-value | | |
|---|---|---|---|---|---|---|---|
| Intervention | | | 0.33 | -0.38~1.04 | 0.3 | | |
| Tai Chi school | | | 0.03 | -0.54~0.58 | 0.91 | | |
| Total intervention times | | | -0.11 | -0.89~0.67 | 0.74 | | |
| subgroup | Type | Amount | WMD (95% CI) | P | heterogeneity test | | Between-group heterogeneity test |
| | | | | | $I^2$ | P | |
| Intervention | Tai Chi alone | 6 | -1.23 [-1.46, -0.99] | < 0.01 | 86% | < 0.1 | $I^2$ = 82.7%, P = 0.003 |
| | Tai Chi combined with other conventional treatment | 3 | -1.02 [-1.29, -0.75] | < 0.01 | 83% | < 0.1 | |
| | Ai Chi | 1 | -0.54 [-0.86, -0.22] | < 0.01 | - | - | |
| Tai Chi school | Chen-style | 3 | -1.37 [-1.90, -0.84] | < 0.01 | 91% | < 0.1 | $I^2$ = 0%, P = 0.47 |
| | Yang-style | 2 | -1.10 [-1.48, -0.73] | < 0.01 | 52% | 0.15 | |
| | others | 5 | -1.00 [-1.26, -0.75] | < 0.01 | 89% | < 0.1 | |
| Total intervention sessions | > 30 | 5 | -1.09 [-1.33, -0.84] | < 0.01 | 85% | < 0.1 | $I^2$ = 0%, P = 0.96 |
| | ≤ 30 | 5 | -1.10 [-1.31, -0.88] | < 0.01 | 79% | 0.46 | |

**Table 3. GRADE quality rating results.**

| Included studies | Risk of bias | Inconsistent evaluation | Indirectness | Inaccuracy | Publishing bias | Tai Chi | Control group | WMD/SMD (95% CI) | Quality results |
|---|---|---|---|---|---|---|---|---|---|
| **1. pain** | | | | | | | | | |
| 12(RCT) | Downgrading one level① | No downgrading | No downgrading | No downgrading | No downgrading | 490 | 396 | *WMD* -1.09 (-1.26, -0.92) | ⊕⊕⊕⊙ medium |
| **2. Disability** | | | | | | | | | |
| 4(RCT) | Downgrading one level② | No downgrading | No downgrading | Downgrading one level③ | No downgrading | 148 | 148 | *SMD* -1.75 (-2.02, -1.48) | ⊕⊕⊙⊙ low |
| **3. body health** | | | | | | | | | |
| 3(RCT) | No downgrading | No downgrading | No downgrading | Downgrading one level③ | No downgrading | 154 | 154 | *WMD* 4.18 (3.41, 4.95) | ⊕⊕⊕⊙ medium |
| **4. mental health** | | | | | | | | | |
| 3(RCT) | No downgrading | No downgrading | No downgrading | Downgrading one level③ | No downgrading | 154 | 154 | *WMD* 3.23 (2.42, 4.04) | ⊕⊕⊕⊙ medium |
| **5. Adverse events** | | | | | | | | | |
| 1(RCT) | Downgrading one level② | No downgrading | No downgrading | Downgrading one level③ | No downgrading | 80 | 80 | *WMD* 1.3 (1.44, 1.16) | ⊕⊕⊙⊙ low |

Confidence Interval; *WMD*: Weighted Mean Difference; *SMD*: Std Mean Difference.

① There was partial deviation from the intended intervention; ② The randomization process was incomplete; ③ The sample size was pretty small.

**3.4.2 Effect of Tai Chi on secondary outcome indicators.** Waist specific disability: Four studies reported on the effect of Tai Chi on the Disability Index in patients with CLBP, among that one using the ODI scale and the other three using the RMDQ scale. The analysis of heterogeneity tests showed no heterogeneity among studies ($p = 0.51$, $I^2 = 0\%$). A fixed effect model was used for analysis, and the results showed that the index of Disability was significantly lower after Tai Chi therapy than in the control group, and the difference was statistically significant [$SMD$ (95% CI) = -1.75 (-2.02, -1.48), $p < 0.01$], as shown in Fig 4(A).

Quality of life: Three studies assessed the effect of Tai Chi on the quality of life of patients with CLBP using the SF-36 questionnaire, which has eight dimensions, of which four dimensions—physical functioning, role physical, body pain and general health—form the physical component summary, while vitality, social functioning, role emotional and mental health form the mental component summary, combining effect sizes for the two sections respectively.

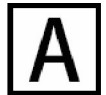

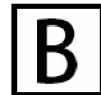

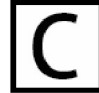

**Fig 4.** (A) Meta-analysis forest plot of the effect of Tai Chi on disability; (B) Meta-analysis forest plot of the effect of Tai Chi on physical health; (C) Meta-analysis forest plot of the effect of Tai Chi on mental health.

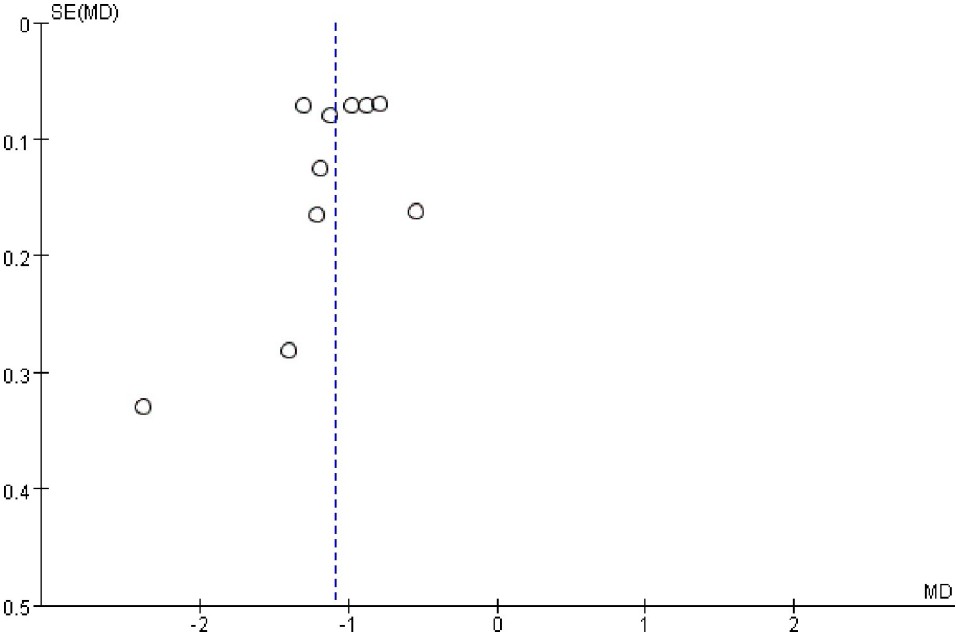

**Fig 5. Funnel plot of pain indicator.**

The analysis of heterogeneity tests showed there was no significant heterogeneity between the two studies ($P = 0.75$, $I^2 = 0\%$; $P = 0.68$, $I^2 = 0\%$). A fixed effect model was used for analysis, and the results showed that the grades of physical health and mental health were both significantly higher than in the control group, and the difference was statistically significant [*WMD* (95%*CI*) = 4.18 (3.41, 4.95), $p < 0.01$; *WMD* (95%CI) = 3.23 (2.42, 4.04), $p < 0.01$], as shown in Fig 4(B) and 4(C).

Adverse events: Only one study [17] reported adverse events, with three subjects(3/80, 3.75%) experiencing a slight increase in pain that resolved after 3 weeks and one subject(1/80, 1.25%) experiencing upper back pain that was relieved by correction of upper limb posture. The results suggest a more reliable safety profile for tai chi alone.

**3.4.3 Bias test.** The 10 included studies were all RCTs and, following a rigorous quality assessment, one funnel plot were drawn for pain indicators where the number of observed indicators in the literature was $\geq 5$, with no significant bias found, as shown in Fig 5. Meanwhile, the results of the Egger bias test also showed that there was no publication bias (P = 0.28). Due to the small number of included trials, no publication bias test was performed for other outcome measures.

## 3.5 Evidence level rating results

According to GRADE, the grade of evidence for the improvement of pain intensity, physical health and mental health of patients was "moderate", that for the improvement of disability of patients was "low", and that for adverse events was "low". The summary results of the evidence are shown in Table 3.

## 4 Discussion

This study aimed to summarize the existing evidence, evaluate the curative effect of Tai Chi on CLBP patients, and analyze the influence of Tai Chi on CLBP effect through intervention

measures, schools and intervention times. Moderate and low-quality evidence showed that Tai Chi can reduce the pain degree of patients with low back pain, improve the lumbar disability, and improve the level of physical and mental health. And as a physical and mental exercise therapy, Tai Chi is a relatively safe choice for patients with low back pain. This result was consistent with the conclusion of previous studies [27]. The review reported that compared with passive control or active control (routine training, core training, etc.), mindfulness exercises including Tai Chi, qigong and yoga showed better effects in reducing pain intensity [28]. Another review also showed that traditional Chinese sports, such as Tai Chi and Baduanjin, may have significantly improved the efficacy of CLBP by combining the substantive components (muscle strength, flexibility and stretching training) recommended by the American College of sports medicine (ACSM) [29].

In the process of Tai Chi exercise, it will be emphasized that the action needs to be slow and controlled, and at the same time, the brain consciousness will be adjusted through breathing and meditation. In a systematic review [12], Wen believed that the reason why physical and mental movement affected pain perception and dysfunction levels may be attributed to changes in the central system. CNLBP patients are usually accompanied by abnormal connections between the structure and function of brain regions. Regular Tai Chi exercise can induce regional structural changes in the precentral gyrus, insular sulcus, and middle frontal sulcus [30]. Similarly, studies have proved that Tai Chi can affect brain waves in brain pain areas (parietal lobe and prefrontal lobe), and enable the brain to process the information related to pain more effectively [31]. Shen et al [32] studied the neurobiological mechanism of pain perception and physical function after 8 weeks of Tai Chi intervention, and the results showed that Tai Chi intervention was moderately and highly correlated with the functional connection of amygdala medial prefrontal cortex. In addition, a randomized controlled experiment [33] found that Tai Chi can reduce pain catastrophization, that is, patients' exaggerated and negative thinking stereotype of pain caused by lumbar discomfort, which to some extent mediated the effect of Tai Chi on the pain degree of CNLBP patients. The mechanism of CLBP is complex, and the therapeutic effect of Tai Chi intervention may also include a variety of responses regulated by sympathetic—parasympathetic nerve, nerve—endocrine—immune network, etc [34]. The results of this study showed that moderate-quality evidence showed that Tai Chi can significantly reduce the pain intensity of patients with low back pain, and its combined effect size was large one. Even different intervention measures of Tai Chi had significant effects. In addition, subgroup analysis showed that underwater Tai Chi had no outstanding advantages compared with Tai Chi and Tai Chi alone as adjuvant therapy. Tai Chi in water requires the deliberate use of abdominal breathing to increase the depth of breathing, combined with soothing and relaxing Tai Chi movements, which can maximize the use of water pressure to improve respiratory function and blood circulation [35]. Tai Chi in water combines the advantages of Tai Chi and hydrotherapy, which has potential synergistic benefits in theory. However, this study does not show its advantages, which may be related to the number of articles on Tai Chi in water.

Some studies have investigated the most popular schools of Tai Chi at present, and the researches on the intervention of Tai Chi on low back pain focused on the Chen and Yang schools [36], which is consistent with the results of the literature we included. The results of this study showed that there was no obvious difference in the curative effect between Yang's and Chen's. Some studies believed that Chen Style Taijiquan was more powerful and flexible than that of Yang's in lower limb activities, and had more obvious effects on lower limb nerve function of the elderly with lower back pain [28]. In fact, Chen's Taijiquan can be divided into da jia and xiao jia, with two routines. The first routine: the twining force is obvious. The action requires paying attention to the movement of waist and abdomen muscles. The characteristics

of arc spiral can fully mobilize the waist muscles; Exercise requires the combination of movement and breathing, which can mobilize the diaphragm, transversus abdominis and other deep core muscle groups. The second routine: Pao Chui is relatively powerful, with strong explosive force, many jumping movements and high exercise intensity, which is not suitable for the elderly and the weak. Compared with Chen's, Yang's Tai Chi is more gentle and relaxed; it is characterized by firmness behind the gentle appearance, with no jumping and no vertical movement. In addition, after learning about other schools of Tai Chi, we found that the He-Style Tai Chi is required to be natural and more coordinated, while Wu's Tai Chi is light in action and sensitive in footwork. Although different schools of Tai Chi have their own characteristics, they all have the same thing that they all require quiet mind and soft body, relaxed and gentle, dynamic and static combination, coordinated and natural, and dominate the waist. When formulating the training content of Tai Chi, selective exercises can be carried out from specific actions according to the state of CLBP patients [37]. Single whip, cloud hand and other moves, as well as moves such as Peng (warding off), Lv (rolling back), Ji (pressing), An (pushing), Kao (body stroke), all of them require a certain degree of stability at the waist. The subgroup results of total interventions times showed that more than 30 times of Tai Chi exercises did not significantly improve the pain intensity of CLBP in patients compared with that of less than 30 times. Previous studies have explored the effects of subject status and different training program elements (total intervention sessions, weekly intervention frequency, and single session duration), and have suggested that age and total practice time may be the main sources of intervention efficacy. And intervention efficacy tended to decrease with age, but long-term practice significantly reduced pain severity [29].

The secondary indicators of this study were disability, quality of life, and adverse events. Among them, the quality of life indicators mainly combined the SF-36 scale with the assessment results of physical function and psychological status related to the quality of life of CLBP patients. This study used GRADE to assess the level of evidence for the results of the study. Low-quality evidence showed that Tai Chi intervention for patients with chronic low back pain resulted in greater effects on disability, physical health, and mental health than the control group. The low quality of the included studies and the small sample size are the main reasons for the low quality of the evidence. In terms of physical exercises, Tai Chi combines multiple forms of training for muscle strength, stability, static and dynamic balance, and these core principles are very similar to core stability training. Tai chi exercise can make the practitioner's muscle strength and bone density value increased significantly, promote its motor function, balance function improvement, so as to effectively improve the symptoms of low back pain [36]. There is no systematic study on the effect of Tai chi on the psychological state of CLBP patients, and its mechanism may be related to the pain relief mechanism. Research has shown that Tai Chi combined with aerobic exercise and meditation to treat clinical patients with negative emotions may be achieved through the regulation of the prefrontal cortex, which plays a pivotal role in regulating human mental health [38]. In the included studies, the reports of adverse events only showed mild pain exacerbation and that the pain disappeared after specific intervention. Tai Chi is a relatively safe rehabilitation exercise with slow and controlled movements. It is important to note that during the practice of Tai Chi, the knee joint is often in a semi-bent state, and the knee may be subjected to high mechanical loads, which poses a risk of injury [39]. Therefore, if patients have insufficient knee stability, they need to pay attention to adjusting the exercise load appropriately.

The strengths of this study are that it included another form of Tai Chi: Ai Chi, based on a comprehensive evaluation of Tai Chi for CLBP relief. At the same time, the influence of different intervention measures, Tai Chi schools, and total intervention sessions on CLBP was explored, which provided further reference for the design of prescriptions such as Tai Chi.

However, there are several limitations to this study. Firstly, 10 of the 4 English and 6 Chinese literature included 9 studies were from China, which may be subject to regional bias. In fact, because Tai Chi is a traditional Chinese fitness modality, it is not a limitation of this study, but rather a general limitation of research on this research theme. Secondly, There is significant heterogeneity in the pooled results of pain indicators, and the source of heterogeneity has not been explored. Thirdly, There are few included studies in some subgroups, and it is expected that more relevant studies will be conducted in the future to further expand the results of this meta-analysis. Finally, the included studies were mostly of small sample size and low quality, and it is recommended that more studies of large sample size and high quality could be included in the future.

## 5 Conclusion

The research found that Tai Chi had a positive effect on relieving pain intensity, functional impairment and quality of life in patients with CLBP. In terms of research design, both Tai Chi used alone and Tai Chi as a complementary therapy had good results; the effect of Ai Chi has not been proven. There was no significant difference between Chen Style Tai Chi and Yang's Tai Chi on the effect of CLBP intervention with both more than 30 times interventions or that of less than 30 times.

## Supporting information

**S1 Checklist. PRISMA 2020 main checklist.**
(DOCX)

**S1 File. Search strategy.**
(DOCX)

**S1 Data. Full-text screening and datasets.**
(XLSX)

## Author Contributions

**Conceptualization:** Hailun Kang, Min Yang, Mengke Li, Qinqin Lin.

**Data curation:** Rui Xi, Qin Sun.

**Formal analysis:** Hailun Kang, Min Yang.

**Funding acquisition:** Qinqin Lin.

**Investigation:** Min Yang, Mengke Li.

**Methodology:** Hailun Kang, Mengke Li, Rui Xi.

**Project administration:** Min Yang.

**Software:** Hailun Kang, Mengke Li.

**Supervision:** Min Yang.

**Validation:** Rui Xi, Qin Sun.

**Visualization:** Hailun Kang, Mengke Li.

**Writing – original draft:** Hailun Kang, Mengke Li.

**Writing – review & editing:** Min Yang.

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
