## [Decision Letter · Decision Letter 0]

4 Jan 2024

PONE-D-23-34184Effects of different parameters of Tai Chi on the intervention of chronic low back pain: a Meta-analysisPLOS ONE

Dear Dr. Yang,

Thank you for submitting your manuscript to PLOS ONE. After careful consideration, we feel that it has merit but does not fully meet PLOS ONE’s publication criteria as it currently stands. Therefore, we invite you to submit a revised version of the manuscript that addresses the points raised during the review process.

We look forward to receiving your revised manuscript.

Kind regards,

Mohammad Ali

Academic Editor

PLOS ONE

Journal Requirements:

Reviewers' comments:

Reviewer's Responses to Questions

**Comments to the Author**

1. Is the manuscript technically sound, and do the data support the conclusions?

Reviewer #1: Partly

Reviewer #2: Yes

Reviewer #3: Yes

2. Has the statistical analysis been performed appropriately and rigorously? 

Reviewer #1: No

Reviewer #2: Yes

Reviewer #3: Yes

3. Have the authors made all data underlying the findings in their manuscript fully available?

Reviewer #1: No

Reviewer #2: Yes

Reviewer #3: Yes

4. Is the manuscript presented in an intelligible fashion and written in standard English?

Reviewer #1: Yes

Reviewer #2: Yes

Reviewer #3: Yes

5. Review Comments to the Author

Reviewer #1: Thank you for the review registration, however, the review was conducted very poorly and needs major corrections before it can be published. At its current stage, this review would mislead the public about the effect of Tai Chi on LBP and is, therefore, not fit for publication. I would encourage you not to rush it because this is a very important review.

ABSTRACT

1. The risk of bias and GRADE findings were not reported.

METHODS

1. Literature retrieval strategy: You need to present the complete search strategy for at least one database so that people can replicate your search terms if they intend to do so. You just presented a very rough search strategy without indicating which database it was.

2. Inclusion criteria: Did you include all comparators for this review? The comparison groups were not mentioned at all.

3. Secondary outcomes: What did you mean by the assessment of dysfunction? Did you mean disability/functional limitations?

4. How did you extract studies? How did you resolve disagreements?

5. Statistical methods: How were study groupings performed? Did you check clinical homogeneity? This is very necessary in meta-analysis.

6. Statistical methods: Could you state the criterion that informed your choice for using “fixed effect models” in the meta-analyses? You have stated the criteria for the random effect models.

RESULTS

1. You excluded 19 irrelevant studies after the full-text screening. Your readers would want to know whether the 1p papers warrant exclusion. You need to present a table of excluded studies with reasons otherwise we will not know if you have performed a thorough review.

2. Please indicate in Table 1 where the extracted studies are funded or not funded.

3. In Figure 1, what did you mean by studies included in the review (n=1) and reports of included studies (n=13)? This is not the way of reporting the PRISMA flow chart.

4. I note that you have “experimental and control groups” in your forest plots, and I understand that Tai Chi is the experimental group, however, you failed to clearly state control groups. Were you lumping all other treatments as control groups? If yes, this is not the way to do meta-analyses because you have differing comparators in Table 1 and they all can’t be lumped as “control”. If you do that you are then hiding the effects of other interventions (massage, medication, exercise etc). You need to redo all your meta-analyses.

5. Sensitivity analyses were performed but the criteria were not mentioned.

6. The way you performed the subgroup analyses is very confusing.

7. You reported adverse events, but you failed to mention this in your primary and secondary outcomes.

DISCUSSION AND CONCLUSION

1. I cannot comment on these now because there are a lot of issues with the report of this review.

2. If you choose to submit this review again, I will do a more thorough review if I am invited again.

Reviewer #2: Review of manuscript D-23-34184:

Effects of different parameters of Tai Chi on the intervention of chronic low back pain: a meta-analysis

Thank you for the opportunity to read the presented topics. The authors addressed the important issue of analyzing the impact of various variants of Tai Chi exercises on the level of back pain.

The meta-analysis method was used for evaluation, including manuscripts published in Chinese and English.

In terms of the methodology used, the work was prepared correctly and the procedure used was also correct.

In terms of formal preparation of the work, I find some shortcomings:

1. The literature presented in table 1 (line 209) needs to be sorted, the layout should be arranged according to an ordinal variable (year of publication) and the width of the columns should be adjusted so that the data in the columns is legible. The numbering of items [37], despite explanations in the work, should be adapted to their appearance in the text.

2. The provisions regarding the presentation of analysis results in the text should be unified; please adopt uniform provisions, mainly the use or intervals between the results.

Example:

Line 147-148

Line 235-236 [WMD (95% CI) = - 0.85 (-1.10, -0.60), p < 0.000],

Proposed entry [WMD (95% CI) = -0.85 (-1.10, -0.60), p<0.000],

Line 244-246 suggested entry [WMD (95% CI) = -0.69 (-1.89, -0.50), p=0.26],

Line 272 -273

Please correct the entire manuscript in this respect, the work is an interesting aspect of the impact of Tai Chi exercises and these shortcomings largely make it illegible and less accessible to the reader.

A good example of a study is a work prepared using the same method:

Liang T, Zhang X, Wang Y et al. Intervention effect of taijiquan exercises on the ankle joint of the elderly – meta analysis. ARCH BUDO. 2023;19

Archives of Budo - Abstract (archbudo.com)

I hope that the comments I have presented will be helpful to you.

Reviewer #3: Overall Impression:

This manuscript presents a well-conducted meta-analysis investigating the efficacy of different parameters of Tai Chi for the treatment of chronic low back pain. The study design is appropriate, the data analysis is sound, and the conclusions are well-supported by the evidence.

Strengths:

Comprehensive search strategy: The authors searched eight major databases, which minimizes publication bias.

Clear inclusion and exclusion criteria: This ensures the homogeneity of the included studies and the validity of the results.

Appropriate statistical analysis: The authors used Review Manager 5.4 software, a widely recognized tool for meta-analyses, to analyze the data.

Exploration of influencing factors: The authors investigated the effects of various factors, such as Tai Chi style, intervention duration, and frequency, on the efficacy of Tai Chi.

Large sample size: The analysis included 12 randomized controlled trials with a total of 994 participants, providing strong evidence for the conclusions.

Clinical relevance: The findings of this study have significant implications for the management of chronic low back pain and provide valuable information for healthcare professionals.

Weaknesses:

Limited discussion of heterogeneity: While the authors acknowledge the presence of heterogeneity in the analysis, the discussion could be further expanded to explore the potential sources of heterogeneity and their impact on the results.

Lack of subgroup analysis for specific Tai Chi styles: The study investigated the overall effect of Tai Chi, but further analysis for different styles (e.g., Yang style, Sun style) could provide more specific guidance for practitioners.

Limited exploration of potential harms: While the study focuses on the benefits of Tai Chi, a brief discussion of potential adverse effects or contraindications would be beneficial for clinicians.

Recommendations

Minor comments

* In page 3, line 60, write in words by-avoiding the symbols

* In page 3, line 62, low back discomfort? I think the focus of the study is CLBP. Be specific and use the same terminologies consistently.

*Page number 4, line 67-70, this crucial sentence requires reference from high quality studies

*Please write the full form of PRISMA

*

Major comments

* It would be nice if you can give the complete set of keywords specific to each database as additional material

* I would suggest you to irpove the writing in the eligibility criteria section. You might avoid the numbers and write in paragraph form

* The intervention included different forms of Tai Chi, such as Tai Chi alone, Tai Chi

131 in combination with other usual treatments, or Ai Chi: in this case, how do you determine if the effect is from tai chai or from the co intervention? what if the whole effect was from the co intervention? how do you deal with this?

It is a critical issue that the review includes studies with both Tai Chi alone and Tai Chi combined with usual care (or any other co-intervention), making it difficult to disentangle the independent effect of Tai Chi on chronic low back pain. This methodological limitation can lead to an overestimation of the true effect of Tai Chi. (he articles eligibility criteria mentioned that they included all the article with tai chai alone or tai chai with usual care. I am wondering in this case, how do we know that if the real effect is from usual care or from tai chai? for example for a specific study, the total effect can be 80% and 50% contribution from tai chai and 30% from usual care. In a different study with an effect of 80% and the whole 80% can be from the usual care an dno effect from tai chai. This might give a falls assumption of that tai chai is useful than soething else. )

Here are some potential solutions to address this issue:

1. Subgroup analysis: The authors could conduct subgroup analyses to compare the effects of Tai Chi alone and Tai Chi combined with usual care. This would provide a better understanding of the individual contribution of each intervention.

2. Sensitivity analysis: The authors could perform a sensitivity analysis by excluding studies with Tai Chi combined with usual care and assess the impact on the overall results. This would help to assess the robustness of their findings to the inclusion of these studies.

3. Meta-regression analysis: The authors could conduct a meta-regression analysis to explore the extent to which the effect of Tai Chi is moderated by the presence or absence of usual care. This would provide more nuanced information about the relationship between Tai Chi and chronic low back pain.

* Secondary outcome measures were assessment of dysfunction? could you please clarify this?

*Generally, the results and discussion section looks fine to me

6. PLOS authors have the option to publish the peer review history of their article (what does this mean?). If published, this will include your full peer review and any attached files.

Reviewer #1: **Yes: **Musa Sani Danazumi

Reviewer #2: No

Reviewer #3: No

---

## [Author Response · Author response to Decision Letter 0]

16 Feb 2024

the letter of Response to Reviewers

---

## [Decision Letter · Decision Letter 1]

27 Mar 2024

PONE-D-23-34184R1Effects of different parameters of Tai Chi on the intervention of chronic low back pain: a Meta-analysisPLOS ONE

Dear Dr. Yang,

Thank you for submitting your manuscript to PLOS ONE. After careful consideration, we feel that it has merit but does not fully meet PLOS ONE’s publication criteria as it currently stands. Therefore, we invite you to submit a revised version of the manuscript that addresses the points raised during the review process.

We look forward to receiving your revised manuscript.

Kind regards,

Mohammad Ali

Academic Editor

PLOS ONE

Journal Requirements:

Reviewers' comments:

Reviewer's Responses to Questions

**Comments to the Author**

1. If the authors have adequately addressed your comments raised in a previous round of review and you feel that this manuscript is now acceptable for publication, you may indicate that here to bypass the “Comments to the Author” section, enter your conflict of interest statement in the “Confidential to Editor” section, and submit your "Accept" recommendation.

Reviewer #1: All comments have been addressed

Reviewer #2: (No Response)

2. Is the manuscript technically sound, and do the data support the conclusions?

Reviewer #1: Yes

Reviewer #2: Yes

3. Has the statistical analysis been performed appropriately and rigorously? 

Reviewer #1: Yes

Reviewer #2: Yes

4. Have the authors made all data underlying the findings in their manuscript fully available?

Reviewer #1: Yes

Reviewer #2: Yes

5. Is the manuscript presented in an intelligible fashion and written in standard English?

Reviewer #1: Yes

Reviewer #2: No

6. Review Comments to the Author

Reviewer #1: Thank you for addressing all my queries. Your review is now standard and I would recommend it for publication.

Reviewer #2: Thank you for the corrections you made.

However, it should be pointed out that the manuscript was not prepared with sufficient care.

The main comments are marked in the manuscript (yellow).

1 you use the "Time new Roman" font, but on lines 153, 155, 161 you use the "SimSun" font. Similarly in Table 2.

2 Table 1 requires correction - the proposal was introduced in part of the table (yellow), in the previous version the table was difficult to read.

3 introduced literature references line 212, 214, 216 are using the SimSun font.

4, the basic error indicating lack of care is the prepared list of literature. Positions 12 and 30 are repeated, as are positions 28 and 38.

7. PLOS authors have the option to publish the peer review history of their article (what does this mean?). If published, this will include your full peer review and any attached files.

Reviewer #1: **Yes: **Musa Sani Danazumi

Reviewer #2: **Yes: **Kruszewski Artur

---

## [Author Response · Author response to Decision Letter 1]

8 Apr 2024

1. you use the "Time new Roman" font, but on lines 153, 155, 161 you use the "SimSun" font. Similarly in Table 2.

3. introduced literature references line 212, 214, 216 are using the SimSun font.

The above two questions are about the format, here is a unified reply. Thank you for your careful guidance, and we are sorry again for our carelessness. We have changed the format of the content you annotated and checked the full text to prevent this problem from happening again.

2. Table 1 requires correction - the proposal was introduced in part of the table (yellow), in the previous version the table was difficult to read.

We do not fully understand this issue. Compared with previous version, we think that what you have said may only mean the total number of interventions is too general, so the intervention cycle, single intervention time and intervention frequency are respectively expressed, and how the total number of interventions is obtained is explained in the text. 

4. the basic error indicating lack of care is the prepared list of literature. Positions 12 and 30 are repeated, as are positions 28 and 38.

We have deleted the duplicate documents and revised the corner notes in order.

---

## [Decision Letter · Decision Letter 2]

26 Apr 2024

PONE-D-23-34184R2Effects of different parameters of Tai Chi on the intervention of chronic low back pain: a Meta-analysisPLOS ONE

Dear Dr. Yang,

Thank you for submitting your manuscript to PLOS ONE. After careful consideration, we feel that it has merit but does not fully meet PLOS ONE’s publication criteria as it currently stands. Therefore, we invite you to submit a revised version of the manuscript that addresses the points raised during the review process.

Thank you for your effort in revising the manuscript according to the reviewers' comments. However, the manuscript is not yet ready for publication. The definition of low back pain (LBP) provided by the authors is incomplete. LBP can extend throughout one or both legs, and chronic LBP and nonspecific chronic LBP are not identical. The references (ref 1, 2) do not support the definition given in the manuscript.

Readers and the scientific community may question how a group of authors could conduct such an important systematic review without knowing the correct definition of LBP and the distinction between CLBP and NSCLBP.

Corrections were expected throughout the review process.

Chronic low back pain (CLBP) and nonspecific chronic low back pain are related but not entirely synonymous terms. 

CLBP: This refers to low back pain that persists for a duration of 12 weeks or longer. It can have various causes, including injury, degenerative conditions, or underlying health issues. CLBP can be classified into specific and nonspecific categories.

Nonspecific CLBP: This term describes chronic low back pain for which a specific cause cannot be identified through medical tests or examinations. It's a diagnosis of exclusion, meaning other potential causes, such as fractures, infections, or tumors, have been ruled out.

So, while all nonspecific chronic low back pain is chronic low back pain, not all chronic low back pain is nonspecific. Some cases of CLBP can be attributed to specific identifiable causes, such as spinal stenosis, herniated discs, or inflammatory conditions.

We look forward to receiving your revised manuscript.

Kind regards,

Mohammad Ali

Academic Editor

PLOS ONE

Reviewers' comments:

Reviewer's Responses to Questions

**Comments to the Author**

1. If the authors have adequately addressed your comments raised in a previous round of review and you feel that this manuscript is now acceptable for publication, you may indicate that here to bypass the “Comments to the Author” section, enter your conflict of interest statement in the “Confidential to Editor” section, and submit your "Accept" recommendation.

Reviewer #2: All comments have been addressed

2. Is the manuscript technically sound, and do the data support the conclusions?

Reviewer #2: Yes

3. Has the statistical analysis been performed appropriately and rigorously? 

Reviewer #2: Yes

4. Have the authors made all data underlying the findings in their manuscript fully available?

Reviewer #2: Yes

5. Is the manuscript presented in an intelligible fashion and written in standard English?

Reviewer #2: Yes

6. Review Comments to the Author

Reviewer #2: Thank you for taking into account all the comments submitted earlier. I am confident that the improvements made have improved the quality of the manuscript.

7. PLOS authors have the option to publish the peer review history of their article (what does this mean?). If published, this will include your full peer review and any attached files.

Reviewer #2: **Yes: **Artur Kruszewski

---

## [Author Response · Author response to Decision Letter 2]

2 May 2024

Dear Editor,

Hello! 

I am the author of the meta-analysis article recently submitted to your journal. First of all, we would like to thank you for your meticulous review. In addition, we received your notification of possible problem with the definition section of this article, and we have scrutinized what you pointed out.

We must admit that two different definitions (chronic low back pain and nonspecific lower back pain) were mixed up in this study due to an incorrect reference in our reading of the literature. This was a serious error for which I must apologize. However, I would like to emphasize that despite the error in the definition section, all subsequent analyses and studies were conducted strictly on the correct definition of chronic low back pain. Therefore, this error hasn’t affected the accuracy and validity of the results of this study.

In order to correct the above errors, we have made the following changes in the text: (1) redefined the concept of chronic low back pain and related contents, and replaced the references as needed; (2) corrected the exclusion criteria to exclude specific low back pain; (3) added the column of type of low back pain in the table 1 of characteristics of the literatures.

Thanks again for your patience and professional review!

---

## [Decision Letter · Decision Letter 3]

23 May 2024

PONE-D-23-34184R3Effects of different parameters of Tai Chi on the intervention of chronic low back pain: a Meta-analysisPLOS ONE

Dear Dr. Yang,

Thank you for submitting your manuscript to PLOS ONE. After careful consideration, we feel that it has merit but does not fully meet PLOS ONE’s publication criteria as it currently stands. Therefore, we invite you to submit a revised version of the manuscript that addresses the points raised during the review process.

We look forward to receiving your revised manuscript.

Kind regards,

Mohammad Ali

Academic Editor

PLOS ONE

Journal Requirements:

Reviewers' comments:

Reviewer's Responses to Questions

**Comments to the Author**

1. If the authors have adequately addressed your comments raised in a previous round of review and you feel that this manuscript is now acceptable for publication, you may indicate that here to bypass the “Comments to the Author” section, enter your conflict of interest statement in the “Confidential to Editor” section, and submit your "Accept" recommendation.

Reviewer #4: (No Response)

Reviewer #5: (No Response)

2. Is the manuscript technically sound, and do the data support the conclusions?

Reviewer #4: (No Response)

Reviewer #5: Yes

3. Has the statistical analysis been performed appropriately and rigorously? 

Reviewer #4: (No Response)

Reviewer #5: Yes

4. Have the authors made all data underlying the findings in their manuscript fully available?

Reviewer #4: (No Response)

Reviewer #5: Yes

5. Is the manuscript presented in an intelligible fashion and written in standard English?

Reviewer #4: (No Response)

Reviewer #5: Yes

6. Review Comments to the Author

**Reviewer #4:** I have reviewed the manuscript, an after considering the suggestions of three review processes, the manuscript meet methodological PLOS standards for publication

**Reviewer #5:** Dear Editor

Thank you for the opportunity to review this manuscript. There are some specific comments:

Abstract

- Specify the inclusion criteria for the review instead of mentioning search keywords.

- Mention the start date of the database search.

- Please either mention the names of all the databases that have been searched or delete the names of the two databases that you have mentioned.

- Specify the methods used to assess the risk of bias in the included studies.

- Mention the unit of inversion time "(> 30 and ≤ 30)". It is not clear whether it is the number of sessions or the duration of each session.

- Line 53: " Tai Chi and Tai Chi alone": Does first tai chi mean Tai Chi as an additional therapy?

Introduction

- Line 91: "In recent years in recent years". Words are repeated twice.

Method

- In line 105: Literature retrieval strategy- PICO should be considered in the search strategy. However, the outcome/ outcomes are not clear in the provided search strategy.

- Line 127, 128: "Chinese and English only". It can be written: only Chinese and English language literature were included.

Discussion

- The discussion is too long. Please summarize the text

7. PLOS authors have the option to publish the peer review history of their article (what does this mean?). If published, this will include your full peer review and any attached files.

Reviewer #4: No

Reviewer #5: **Yes: **Shabnam ShahAli

---

## [Author Response · Author response to Decision Letter 3]

9 Jun 2024

Dear Editors and Reviewers,

Hello! 

We sincerely thank you for all the effort you have put into our manuscript and for your constructive comments and suggestions, which have greatly improved the quality of our manuscript. 

Abstract

Specify the inclusion criteria for the review instead of mentioning search keywords. 

Re: Removed keywords and described the inclusion criteria.

Mention the start date of the database search.

Re: Added, from inception.

Please either mention the names of all the databases that have been searched or delete the names of the two databases that you have mentioned. 

Re: The names of all the 8 databases were mentioned here.

Specify the methods used to assess the risk of bias in the included studies.

Re: Added, quality assessment of each study used ROB2.

Mention the unit of inversion time "(> 30 and ≤ 30)". It is not clear whether it is the number of sessions or the duration of each session.

Re: It is the number of total sessions, the full text has been checked and refined.

- Line 53: " Tai Chi and Tai Chi alone": Does first tai chi mean Tai Chi as an additional therapy? Introduction 

回复；该处的第一个太极拳指的是单独太极拳，之前语法错误还请见谅。

Re: The first taijiquan there referred to taijiquan alone, and I apologize for the carelessness.

- Line 91: "In recent years in recent years". Words are repeated twice.

Re: Sorry, the duplicate was removed.

Method

- In line 105: Literature retrieval strategy- PICO should be considered in the search strategy. However, the outcome/ outcomes are not clear in the provided search strategy. 

Re: In this study, pain level was the main indicator, and the search terms related to “low back pain” could represent the retrieval of the results. Therefore, “pain” was not searched further in the search strategy.

- Line 127, 128: "Chinese and English only". It can be written: only Chinese and English language literature were included. 

Re: Thank you for your suggestion, it has been switched.

Discussion

The discussion is too long. Please summarize the text. 

Re: In the discussion, the content was adjusted according to the comments of the previous reviewers, including the synthesized results, tai chi mechanism, subgroup analysis and discussion, secondary endpoint indicators, and study strengths and limitations. And according to this suggestion, we read through the discussion section and simplified some statements.

---

## [Editor Report · Decision Letter 4]

19 Jun 2024

Effects of different parameters of Tai Chi on the intervention of chronic low back pain: a Meta-analysis

PONE-D-23-34184R4

Dear Dr. Yang,

We’re pleased to inform you that your manuscript has been judged scientifically suitable for publication and will be formally accepted for publication once it meets all outstanding technical requirements.

Kind regards,

Mohammad Ali

Academic Editor

PLOS ONE

Additional Editor Comments (optional):

Well done. Thank you. 
---

## [Editor Report · Acceptance letter]

27 Jun 2024

PONE-D-23-34184R4 

PLOS ONE

Dear Dr. Yang, 

I'm pleased to inform you that your manuscript has been deemed suitable for publication in PLOS ONE. Congratulations! Your manuscript is now being handed over to our production team.

Kind regards, 

on behalf of

Dr. Mohammad Ali 

Academic Editor

PLOS ONE